# Bacteriophages for the Treatment of Graft Infections in Cardiovascular Medicine

**DOI:** 10.3390/antibiotics10121446

**Published:** 2021-11-25

**Authors:** Simon Junghans, Sebastian V. Rojas, Romy Skusa, Anja Püschel, Eberhard Grambow, Juliane Kohlen, Philipp Warnke, Jan Gummert, Justus Gross

**Affiliations:** 1G. Pohl-Boskamp GmbH & Co. KG, 25551 Hohenlockstedt, Germany; s-junghans@web.de; 2Department of Cardio-Thoracic Surgery, Heart and Diabetes Centre NRW, University Hospital of the Ruhr-University Bochum, 32545 Bad Oeynhausen, Germany; srojashernandez@hdz-nrw.de (S.V.R.); jgummert@hdz-nrw.de (J.G.); 3Department for General, Visceral, Thoracic, Vascular and Transplantation Surgery, Rostock University Medical Center, 18057 Rostock, Germany; romy.skusa@med.uni-rostock.de (R.S.); anja.pueschel@med.uni-rostock.de (A.P.); Eberhard.grambow@med.uni-rostock.de (E.G.); Juliane.kohlen@uni-rostock.de (J.K.); 4Institute for Medical Microbiology, Virology and Hygiene, University Medicine Rostock, 18057 Rostock, Germany; PhilippChristoph.Warnke@med.uni-rostock.de

**Keywords:** bacteriophage, phage, infection, prosthesis, vascular graft, antimicrobial resistance

## Abstract

Bacterial infections of vascular grafts represent a major burden in cardiovascular medicine, which is related to an increase in morbidity and mortality. Different factors that are associated with this medical field such as patient frailty, biofilm formation, or immunosuppression negatively influence antibiotic treatment, inhibiting therapy success. Thus, further treatment strategies are required. Bacteriophage antibacterial properties were discovered 100 years ago, but the focus on antibiotics in Western medicine since the mid-20th century slowed the further development of bacteriophage therapy. Therefore, the experience and knowledge gained until then in bacteriophage mechanisms of action, handling, clinical uses, and limitations were largely lost. However, the parallel emergence of antimicrobial resistance and individualized medicine has provoked a radical reassessment of this approach and cardiovascular surgery is one area in which phages may play an important role to cope with this new scenario. In this context, bacteriophages might be applicable for both prophylactic and therapeutic use, serving as a stand-alone therapy or in combination with antibiotics. From another perspective, standardization of phage application is also required. The ideal surgical bacteriophage application method should be less invasive, enabling highly localized concentrations, and limiting bacteriophage distribution to the infection site during a prolonged time lapse. This review describes the latest reports of phage therapy in cardiovascular surgery and discusses options for their use in implant and vascular graft infections.

## 1. Introduction

Both open and endovascular surgical cardiovascular repairs carry the risk of nosocomial infection and a prosthetic vascular graft infection is generally tedious, related to other complications, and often acutely life-threatening [1]. The prevalence of nosocomial infections in surgical patients ranges from 4% to 15% in patients requiring intensive care treatment [2]. The standard treatment of resection and autologous vascular reconstruction can pose a prohibitively high risk in compromised patients, and alternatives are urgently required [3]. Bacteriophages are an especially interesting alternative in the context of the increasing antimicrobial drug resistance and are currently under investigation [4].

## 2. Pathogenic Bacteria and Vascular Graft Infections

Prosthetic vascular graft infection is one of the most feared complications in cardiovascular surgery and is associated with significant morbidity, mortality, as well as increased hospitalization costs [3]. The most common pathogenic bacteria are *Staphylococcus aureus*, *S. epidermidis*, other coagulase-negative staphylococci, *Escherichia coli*, other *Enterobacterales*, *Pseudomonas aeruginosa*, and corynebacteria [5].Bacteria often increase their virulence by attaching themselves to prosthetic material, protecting themselves from the local immune response with the help of biofilms that inhibit phagocytosis. After surface adhesion, bacteria may secrete insoluble gelatinous extrapolymers contributing to biofilm formation. Pathological biofilm formation on biomedical devices results in profound consequences for the patient. Microorganisms growing as biofilm are significantly less sensitive to antibiotics and host defense, as compared with the planktonic growing form of the same microorganisms. Other clinical challenges posed by biofilm infections include uncultivable pathogens, impaired wound healing, chronic inflammation, and the spread of infectious emboli [6,7,8,9]. Next to the treatment of acute infections, prophylactic measures that inhibit the emergence of infections are also important: the prophylactic pre-soaking of vascular grafts in high concentrations of rifampin, for example [10]. Further studies are necessary to determine the influence of rifampin on the restoration of vessel functionality versus its prophylactic effect against vascular graft infections (VGIs). Effective perioperative local delivery of antibacterial substances is crucial for patients with implanted medical devices, especially if implant surgery caused deep wounds. However, prolonged local use of antibiotics is often not possible or simply inadequate. The development of multi-resistant strains against antibiotics represents an additional, serious obstacle to the control of prosthetic infections.

## 3. Infections in Ventricular Assist Device (VAD) Therapy

Infections represent an important complication in long-term mechanical circulatory support (MCS) [11]. System infections are a common complication in VAD patients. Often, these are caused by ascending driveline infections that are mostly Staphylococcus dominated. Reported incidence rates vary between 13% and 80% depending on the definition [12]. In addition to the transcutaneously ascending infection, bacteria can also colonize a VAD system via the bloodstream in the sense of system endocarditis. The diagnosis of a systemic infection is difficult because a positive blood culture is inconclusive, echocardiography cannot be reliable due to artifacts, and positron emission tomography, in combination with computed tomography (PET-CT), can often be false positive [13]. Despite the constant development in the field of left ventricular assist devices (VADs), infection rates have remained constant during the past decade [14]. VAD-related infections can involve intra-thoracic pump components such as pump housing, outflow–graft prosthesis, or driveline. One prospective study demonstrated that 22% of patients who received VADs and developed VAD-associated infections, increasing the one-year mortality five- to sixfold [15]. The most common infection site is the extra-thoracic driveline, which can be affected in up to 50% of ongoing LVAD patients. As previously mentioned, pre-implant management including patient selection, anti-infective therapy, and dental sanitation are important prophylactic measures that reduce the risk of perioperative device infection. When a VAD-associated infection is suspected, thorough assessment by PET CT scan and bacterial cultures is recommended [16]. Depending on the severity of infection, current therapeutical strategies range from antibiotic therapy to surgical debridement, driveline relocation, pump exchange, prolonged antibiotic treatment, or heart transplant [17]. Among these different options, heart transplant involves the lowest reinfection risk despite immunosuppression, because it is the only option that allows the removal of all contaminated device components [18]. However, in a growing number of destination therapy patients, this option is unviable. Therefore, in this collective, device infections often become chronic, thus a therapy-limiting factor.

## 4. Endocarditis in Patients with Prosthetic Heart Valves (PHV)

Infective endocarditis (IE) is a disease with increased morbidity and high in-hospital mortality up to 20% [19]. Despite important advances in prevention, diagnosis, and treatment, IE is among the five deadliest infection syndromes. Lately, its epidemiological profile has shifted from streptococcal infections toward *S. aureus*-related infections. This phenomenon is partially explained by an increasing amount of healthcare contacts (up to 30% of all IE cases are healthcare-associated) and also reflected by changing patient characteristics toward an older collective, less rheumatic heart disease, with a higher portion of a prosthetic heart valve (PHV) or other implantable devices [20]. In a retrospective study from a Danish registry, the cumulative risk of IE was found to be 4.5% for PHV patients in the first 10 years after surgery, which was significantly higher, compared with matched controls without PHV [21]. Patients with IE and PHV have higher mortality, but also a higher incidence of complications, compared with patients with identical infectious profiles but native valves.

The emergence of transcatheter aortic valve replacement (TAVR) for the treatment of severe aortic valve stenosis in high-risk patients has changed the surgical field. First reports on long-term data regarding the incidence of IE have been published recently, demonstrating similar results (5-year incidence of 5.8%), compared with patients undergoing surgical aortic valve replacement (SAVR) [22].

A meta-analysis and systematic review of infective endocarditis after transcatheter aortic valve repair showed 38% overall mortality [23]. Autologous vascular grafts offer a natural resistance to infection, but, even with silver or antibiotic coatings, the incidence of Dacron graft infection is reported to be between 0.5% and 6% (depending on the operation, location, and stage of atherosclerosis), with up to 75% mortality and up to 70% limb amputation rates [3,24]. Endovascular techniques have improved but not eliminated the risk, with a reported rate of 0.5% to 1% [25].

Predisposing factors for the development of a prosthetic vascular graft infection are preoperative hospitalizations duration of operation, amount of blood loss, redo, and extensive access surgery; acute ischemia is also associated with an increased incidence [26]. There is also a causal relationship between existing skin and wound infections and their secondary occurrence as a prosthesis infection, demonstrated by a similar pathogen population in both infection areas. The groin region shows a high incidence of infections (up to 7%), presumably due to anatomical proximity to the anogenital region, the lymph node configuration, and the mechanical stress on the wound during movement in the hip joint [5,27,28,29,30].

It is assumed that around 50–65% of prosthesis infections result from bacterial contamination during surgery [5,27,29,31]. A general distinction is made between early (up to 30 days postoperatively) and late infections, although the classification is arbitrary [5,31,32]. Early prosthesis infections are often assumed to be due to intraoperative contamination and late infections to hematogenous bacterial spread, but the evidence is limited. Late infections are usually caused by insufficient tissue integration of the prosthesis into the graft bed. In addition to *E. coli* and corynebacteria, *S. epidermidis* and other coagulase-negative staphylococci are common [5,31].

Reports of vascular prosthesis replacement using silver-impregnated or rifampicin-soaked prostheses are limited and heterogeneous; uncoated or untreated prostheses show significantly poorer reinfection results and should not be used for open reconstructions [33,34,35]. If a prosthesis is shown to be infected, in situ reconstruction with autologous transplants should be sought after complete removal of the infected material, including rehabilitation of the anatomically adjacent organ segment. An extra-anatomical bypass might be an option but can be difficult or even impossible if there is a high surgical risk. If there is no autologous vascular material or if the focus of the infection is difficult to access—for example, after thoracic endovascular aortic repair (TEVAR), TAVR, or a VAD—it may be feasible only to leave the prosthetic material and employ drainage, irrigation, and vacuum therapy [36]. Systemic antibiotics alone are not sufficient but are nevertheless necessary as concomitant therapy. Initially, if the pathogen is unknown, broad-spectrum antibiotics should be used, and then a pathogen-specific adjustment should be made according to the antibiotic susceptibility pattern [37].

## 5. Bacteriophage Therapy

Bacteriophages (or simply “phages”; from the Greek, “bacteria eater”) are viruses that selectively infect bacterial cells and were first described in 1917 by the French Canadian Félix Hubert d’Hérelle [38]. They are quite stable in the environment and contribute significantly to the regulation of global bacterial mass. In principle, a bacteriophage can be found wherever its corresponding bacterium is but can only multiply where the host is. They are specific and almost always only affect strains within one bacterial species, rarely crossing species boundaries [39]. In the lytic cycle of viral reproduction, phages kill their corresponding bacteria through lysis: once infected, the bacterium host cell then starts the process of reproduction, the destruction of the bacterium, and the release of new phage particles; this process is controlled by enzymes and interaction of bacterial and phage genes. This cycle is typical for virulent phages. In the lysogenic cycle, the bacteriophage nucleic acid is integrated into the host bacterium’s genome or forms a circular replicon in the bacterial cytoplasm and is thus passed on to the next generation of bacteria as a prophage (temperate phage) (Figure 1) [40]. In particular, the lysogenic cycle is of interest in Shiga toxin-producing *E. coli*. As bacteriophages containing the *stx* (Shiga toxin) gene are capable of infecting *E. coli* and perhaps some other related bacteria, this virulence trait can easily be transferred horizontally in a process termed lysogenic conversion [41]. Furthermore, the lytic cycle of these prophages is known to be triggered by DNA damage [42] as well as by various antibiotics [43]. Another significant example of a phage-encoded virulence factor is the cholera toxin. The corresponding lysogenic phage can convert their nonpathogenic host *Vibrio cholerae* through a pathogenic strain by phage conversion [44,45]. In contrast to temperate or lysogenic phages, which do not usually kill the host bacterial cells, the virulent or lytic phages can kill the host cells and should therefore be chosen for phage therapy.

The use of bacteriophages in the Western hemisphere stalled after the widespread introduction of antibiotics following World War II, while the Soviet bloc continued to rely on bacteriophages. The rapid development of highly effective antibiotics was so successful in the treatment of bacterial infections that there was considerable optimism at that time that infectious diseases could be completely eradicated. Numerous new active ingredients such as penicillin, tetracyclines, cephalosporins, macrolides emerged, and there was no cause for concern about antibiotic resistance. Although the first resistant bacterial strains were soon reported, the portfolio of the pharmaceutical industry appeared to be reliably large. The rational handling of these substances was just as neglected as the observance of important hygiene rules, the consistent application of which can effectively prevent the spread of pathogens in the hospital environment [46].

In the mid-1990s, alarming epidemiological data in Germany and many other countries led to a radical rethink in dealing with clinically relevant microorganisms due to the appearance of multi-resistant pathogens. In the lead, the bacterium *S. aureus* showed an increasing degree of insensitivity to common penicillin preparations. To date, considerable public health efforts are required to control the spread of methicillin-resistant *S. aureus* (MRSA) strains, which are resistant to a large number of antibiotics. Worryingly, many microorganisms are increasingly showing resistance to classes of antibiotics that were described as particularly effective in the past. This includes pathogenic bacteria that are among the most common causes of infection in clinical medicine: *E. coli*, *Klebsiella pneumoniae*, *P. aeruginosa,* and *Acinetobacter baumannii*. In contrast to *S. aureus*, it is not possible to free patients who are colonized with these Gram-negative rods from the microorganisms by means of a specific eradication procedure. Due to the large number of genetically determined resistance mechanisms that have now been described, it is not possible in all cases to provide effective antimicrobial drugs. Bacterial strains that are resistant to most, in some cases even to all, available antibiotics are being detected more and more frequently. In Europe, an estimated 33,000 people currently die annually from antibiotic-resistant infections, and, globally by 2050, 10 million people will die annually in the absence of therapeutic alternatives [47,48].

Bacteriophages are dynamic biological agents that multiply in the host bacteria and place different demands on clinical studies and complicate regulatory approval processes more suited to passive drugs such as antibiotics. Although the therapeutical use of phages is common in the countries of the former Soviet bloc, there is a lack of randomized, placebo-controlled, and double-blind studies that would provide scientifically usable data to allow regulatory approval in the Western hemisphere. An increasing number of case reports describe successful phage therapy to treat life-threatening infections [49,50]. However, although some clinical studies have been undertaken, most failed to provide clear evidence of the effectiveness of phage therapy: one study assessing a phage cocktail against *E. coli* and *P. aeruginosa* infections in burn wounds was terminated early due to insufficient effectiveness [51]. This points to the particular challenges facing phages in traditional clinical trials: phage preparations should be individualized and patient specific, but clinical trials require standardization.

To address that, the Phage4Cure project established a possible approval pathway for phage preparations along with a platform for the purification of phages to treat chronic airway infection [52]. However, in the meantime, the clinical application of bacteriophages is limited to last resort, life-threatening situations when no other medically common alternatives for treating the patient are available and “if, in the doctor’s judgment, it gives hope to save life, restore health or alleviate suffering” [53].

Reports of phage treatment are largely in the areas of traumatic osteomyelitis, orthopedic endoprosthesis infection, and superficial wound infections, as well as cystic fibrosis and mucosal viscidosis via inhalation. Reports of phages in cardiovascular surgery are limited. Rubalski and Haverich reported the cases of 8 immunosuppressed patients aged between 13 and 66 years with multi-resistant, *S. aureus*, *E. faecium*, *P. aeruginosa*, *K. pneumoniae,* and *E. coli* infections after organ transplantation or cardiovascular implant [54]. Potentially suitable bacteriophage strains were selected from the well-characterized collection housed in the Gabrichevsky Institute (Moscow, Russia). Lysis efficacy was evaluated by serial dilution spot testing and the efficiency of plating was analyzed by a double layer plaque assay. Individualized phage treatments were applied locally, orally, or by inhalation along with conventional antibiotics and resulted in complete eradication in seven of eight patients without serious side effects.

Graft infections with *P. aeruginosa* are problematic because biofilm formation frequently leads to antibiotic resistance. To examine potential synergy or additive effects and exclude concerns of phage/antibiotic antagonism prior to emergency application of phage OMKO1, we performed in vitro biofilm reduction assays using the strain isolated from the fistular discharge of the patient. Biofilms were grown on 3 mm × 3 mm sections of Dacron by inoculating each section in 150 µL 0.1 × LB broth in a 96-well dish with 50 µL of an overnight culture of *P. aeruginosa* isolated from the fistular discharge of our patient. Chan et al. reported a chronic *P. aeruginosa* infection of a Dacron aortic graft with aortic cutaneous fistula that resolved after a single application of the phages OMKO1 with ceftazidime [55].

In the case of a 41-year-old Marfan patient with an infected fistula between a carotid-subclavian bypass and defibrillator implant, bacteriophage therapy was successful after exhausting conventional options [56]. During surgical debridement, no antiseptics were used. Bacteriophages were applied as a loading dose of bacteriophage solution (10 mL) containing a 1:1 mixture of PYO bacteriophage at $10^{6}$ plaque-forming units (PFU)/mL (R-022600) and staphylococcal bacteriophage Sb-1at $10^{7}$ PFU/mL (R-022876), both from ELIAVA Institute of Bacteriophage, Tbilisi, Georgia. After surgery, a drain was placed into the surgical site before wound closure, through which 5 mL of bacteriophage solution was instilled every 8 h for 14 days. After 21 days, intravenous antibiotics were switched to oral Levofloxacin (500 mg/12 h) and Rifampicin (450 mg/12 h) for 3 months. Follow-up PET-computed tomography (CT) showed a clear decrease in inflammation parameters and the bacteria-specific formation of a biofilm that resulted in resistance to systemic antibiotic treatment. The authors concluded that bacteriophages penetrate and destroy freely circulating bacteria but also their matrix and biofilms. Since the phage’s replication is stopped as soon as the bacterium is destroyed, the risk of side effects is reduced.

Aslam et al. used bacteriophage therapy as an adjunct to intravenous antibiotics in 2 cases of refractory ventricular assist device infections [57]. The bacterial isolates of methicillin-susceptible *S. aureus* (MSSA) were determined, and bacteriophages were identified in vitro for intravenous injection. Case 1 had negative sternal wound cultures for MSSA at the end of therapy. Case 2 developed *P. aeruginosa* bacteremia, which was successfully treated with an adaptation of the antibiotics. In this case, serum neutralizing activity occurred during therapy and reduced viable phage titers in vitro. There were no safety issues, and each patient subsequently received successful heart transplants.

In the case of a patient with mechanical mitral valve replacement and VAD who presented after 4.5 years with fever and elevated inflammation markers out, CT showed an abscess around the pump; culture samples were positive for MSSA [58]. After debridement and evidence of granulation tissue, bacteriophages were applied topically during skin closure. The bacteriophage solution (10 mL) was a 1:1 mixture of PYO bacteriophage with $10^{6}$ plaque-forming units/mL (Georgian pharmaceutical product registration number: R-022600) and staphylococcal bacteriophage Sb-1 with $10^{7}$ plaque-forming units/mL (R-022876), both obtained from the Eliava Institute of Bacteriophage, Microbiology, and Virology, Tbilisi, GA. Bacteriophage products are a mixture of sterile filtrates of phage lysates, which have previously been shown to be active against *S. aureus* isolates from the patient. The surgical site was drained by applying bacteriophage (5 mL) every 8 h for 10 days along with intravenous antibiotics (meropenem and fosfomycin), followed by oral antibiotics (ciprofloxacin/rifampicin): no local antiseptics were used. The patient experienced mild nausea but no other adverse events. The wound healed and the patient was discharged after 16 days. The surgical site showed no local signs of infection at follow-up examination 9 months later. Appendix A provides an overview of case studies using bacteriophages.

## 6. Challenges to Phage Therapy

Bacteriophage therapy requires an exact determination of the pathogenic bacteria meaning at least two interventions are generally necessary: One to take a sample from the focus of infection and a second to introduce the appropriate phage selection after the bacteria mix has been determined. The time required for this process is not always available to the patient who is in a high-risk, life-threatening situation, or the physician who must navigate a complex regulatory and logistical environment. Prophylactic bacteriophages’ use, in turn, is complicated by the galenics required to formulate and deliver the so-called phage cocktail to cover a broad spectrum of pathogens.

Bacteriophage adoption is further complicated by a short half-life and macrophatic degradation in the body, meaning that systemic administration is probably not feasible due to insufficient tissue saturation. In the case of vascular graft infection, for example, intravenous administration would not guarantee adequate transluminal surface contact with the infected prosthesis: contact time might be too short (due to the blood flow rate), and adhesion is further complicated by the formation of biofilms and the formation of a neointima. As a result, phage concentrations with sufficient antibacterial effect would not accumulate in the infected periprosthetic tissue. A topical application would allow sufficient concentration but is limited by the ability to reach the infection location.

Only phage solutions and lyophilized preparations are currently available, so topical application is difficult due to the adhesive and flow dynamics. Even if phages are successfully delivered to the site of infection, they can run into and onto the surrounding tissue, particularly in the case of circular application, such as infected anastomoses. Although this does not normally present a risk of local or systemic adverse reactions, it can compromise efficacy. Preservation of bacteriophages on prosthetic surfaces, for instance, is not known. It is essential that phages remain in the place of application for sufficient time to ensure phage and bacterium interaction; semi-solid and solid formulations are needed.

New galenic formulations need to guarantee the stability of phage preparations. Storage should be in a formulation in which activity is impaired without a significant drop in phage titer during processing and long-term storage: spray-drying, freeze-drying, extrusion drop, emulsion, and polymerization techniques have each been described [51,59,60]. However, phage stability can vary in different formulations (e.g., liquids, gels, powders) and can even vary among different phages [60,61,62]. Haverich et al. demonstrated the biocompatibility of fibrin glue to apply the bacteriophage PA5 in the treatment of the biofilm-forming bacterium *P. aeruginosa* [63]. The possible spontaneous occurrence of mutations in phage stocks that have been stored for a long time or that had accumulated during collection and production must also be considered [64,65].

The body’s immune response to applied bacteriophages is still unclear. The development of phage-specific antibodies (adaptive immunity) showed, on the one hand, a reduction in the concentration of circulating active phages; on the other hand, studies did not show any effects of the immune response on the therapeutic potency of bacteriophages. The immune response plays a decisive role in phage clearance or inactivation, which inevitably results in a reduced effective phage concentration at the infection site [66].

Despite these challenges, the use of bacteriophages to treat vascular prosthetic graft infection is a possible and relatively inexpensive alternative to systemic antibiotic therapy. In the minimally invasive and endovascular surgical fields, bacteriophage coating (comparable to drug-eluting stents or balloons) is feasible. In the case of aortic valve repair, bacteriophages could be bound to the outside of a new valve to subsequently carry out a valve-in-valve procedure with an anti-infective component.

Endografts can be used to deliver treatment to infected parts of the vasculature. For technical reasons, some stent-graft delivery systems have a second inner sheath, which can be used as a phage reservoir; a phage-coated stent-graft would be protected by this inner sheath before deployment (Figure 2) [67]. Questions remain about how shear forces on the surface of the device (often coated with neointima or biofilm) will affect phage adhesion; some decrease in the concentration of bacteriophages should be assumed.

Vacuum-assisted closure therapy is an established method for wound conditioning in superficial infections; vascular graft infections could also be accessible laparoscopically or with robot assistance to allow site exploration, sampling for microbial analyzes and determination, local debridement but also insertion of an endo-foam or gauze prepared with bacteriophages (Figure 3).

There are, therefore, several possibilities to allow bacteriophages to be placed in a minimally invasive manner, in sufficient concentration, and using mechanisms to ensure stable and stationary application at the site of the infection.

## 7. Conclusions

The focus on antibiotics in Western medicine since the mid-20th century slowed the further development of bacteriophage therapy and the experience and knowledge gained until then in their mechanisms of action, handling, clinical uses, and limitations were largely lost. The parallel emergence of antimicrobial resistance and individualized medicine has prompted a radical reassessment of this approach, and cardiovascular surgery is one area in which phages may have an important role in both of these new scenarios. In addition, the lack of new antibiotics has complicated the treatment of bacterial infections—especially infected vascular grafts, where many pathogenic bacteria tend to form a biofilm— has prompted a reevaluation of bacteriophages.

Only a small number of countries—principally Georgia, Poland, and Russia—have used phages for therapeutic purposes and developed specialized research and treatment centers. The safety and effectiveness of bacteriophages for the treatment of infections are documented in these countries, but the adoption of phage therapy in Western Europe is limited due to the lack of a suitable legal and regulatory framework.

As a first step to develop the prophylactic and therapeutical use of bacteriophages in cardiovascular medicine (stand alone or in combination with antibiotics), standardization of phage formulations and laboratory testing is required. There are several possibilities to allow bacteriophages to be placed in a minimally invasive manner, in sufficient concentration, and using mechanisms to ensure stable and stationary application at the site of the infection.

## Figures and Tables

**Figure 1 antibiotics-10-01446-f001:**
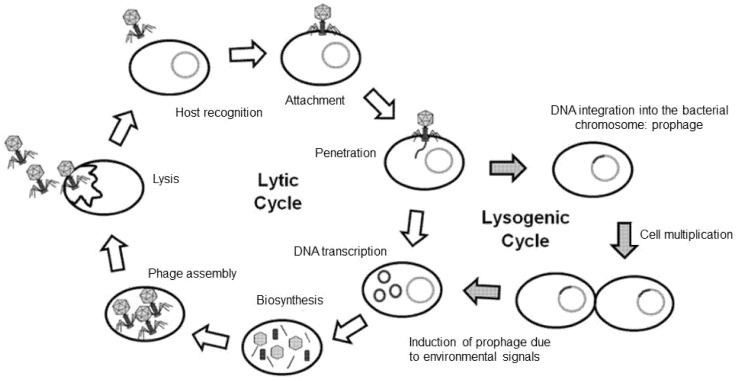
The phage life cycle. Lytic or virulent phages undergo the lytic cycle, in which the host is lysed, and progeny phages are released into the environment. Temperate phages can undergo the lytic or the lysogenic cycle. In the lysogenic cycle, the phage genome is incorporated into the host genome; this phage DNA—now called a prophage—can be induced, leading to the expression of phage DNA and the lytic cycle. Figure adapted from Figure 2 in [40].

**Figure 2 antibiotics-10-01446-f002:**
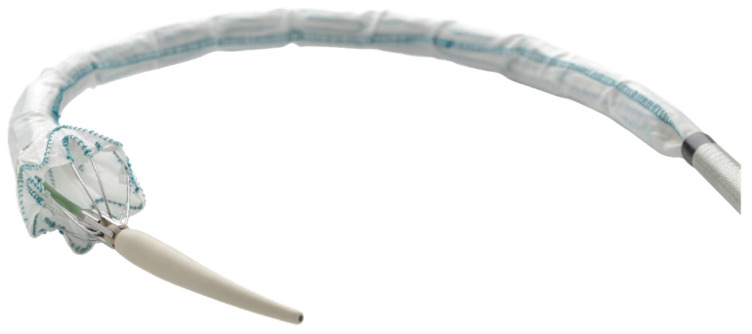
The Relay thoracic stent graft (Terumo Aortic) has a dual-sheath delivery system: a coiled outer sheath advances to the abdomen and a second, flexible inner sheath that is amenable to soaking in bacteriophages.

**Figure 3 antibiotics-10-01446-f003:**
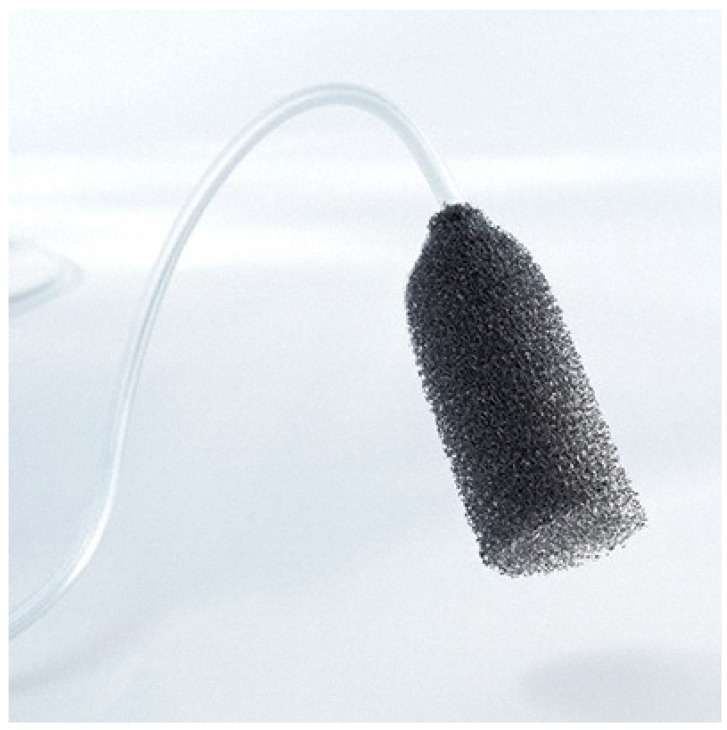
Eso-SPONGE (*B. Braun*) is an endoluminal vacuum therapy device, allowing minimally invasive treatment and prevention of anastomotic leakages.

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
