# Peer review of "Bacteriophages for the Treatment of Graft Infections in Cardiovascular Medicine"

_antibiotics, 2021, doi:10.3390/antibiotics10121446_

Round 1

Reviewer 1 Report

Manuscript entitled "Bacteriophages for the treatment of graft infections in cardiovascular medicine" by authors Simon Junghans et al is reviewing the use of bacteriophages for the treatment. Though the review is good, authors needs to address the following

  1. Line#54: VGIs - please abbreviate
  2. Line#137: Bacteriophage therapy: please provide illustrating  figure. This will make readers to understand easily (like life cycle indicating the mechanism of action at various stages).
  3. Was there any animal studies reported to test the phages safety?. Please list them.
  4. Please provide at least one table with type of phages used in different disease conditions with references then narrow down to cardiovascular disease. 

It will be good if the review has more figures and tables to easily understand.

Author Response

  1. The abbreviation “VGI = Vascular graft infections” was added (in line 61 and in list of abbreviations).

  1. Illustrating figure was added. Please refer to Figure 3.

  1. In the case reports cited, no preoperative animal experiments were carried out to test the safety of bacteriophages. This is understandable since it is assumed that bacteriophages are natural, ubiquitous and symbiotic microorganisms. The areas of application in cardiovascular medicine are usually urgent interventions that threaten the patient's life. An animal experiment would practically not be conceivable in terms of time, furthermore not necessary in terms of indication for use as a healing attempt according to §37 of the Helsinki Declaration. In contrast, a preoperative in vitro sensitivity test to assess the effectiveness of a phage application should be mandatory.

  1. The following table provides an overview of case studies using bacteriophages.

“Table 1. Summary of reported case studies using phage therapy.”

Reviewer 2 Report

The subject of this review is important, as phage therapy is considered to be one of possible alternative therapies of infections in the era of the antibiotic resistance crizis. Definitely, such a paper is worth publishing, however, I recommend major revision before acceptance of this paper. The most important limitation of this manuscript is marginalization of the microbiological aspects of the reviewed subject. More detailed description of various aspects in this field is required. Specific critical points which should be addressed are listed below.

Major points:

  1. Lines 46-48. The problem of biofilm formation is crucial in the field of clinical microbiology. Therefore, description of the phenomenon of biofim formation, and properties of bacterial biofilms deserve significantly more detailed description than just mentioning. At least one paragraph should be devoted to these aspects.
  2. Section 3 (lines 60-79). Most common etiological factors of VAD infections should be mentioned and discussed briefly.
  3. Lines 144-149. Life cycles of bacteriophages are presented very superficially. This subject requires significantly more detailed description. For instance, indication that bacteriophage development is controlled by enzymes, not even mentioning what kinds of enzymes, is an example of a negligent description of an important process. I strongly recommend extending this part of the manuscript by at least an additional paragraph. A scheme of lytic and lysogenic phage development would be very useful. Moreover, the process of prophage induction should be described as an important medical problem (consider to mention pathogenicity of Shiga toxin producing Escherichia coli, Vibrio cholerae virulence factors and others). Then, it should be clearly stated that virulent, rather than temperate, bacteriophages should be chosen for phage therapy.
  4. Lines 198-236. The cases are described very generally. At least more details about phage doses and conditions of treatments should be indicated.

Minor points:

  1. Lines 45-46. Please, check microbiological nomenclature, and use proper names of baterial species and other taxonomic groups. Note also that Escherichia coli belongs to Enterobacterales, thus, it is not proper to mention both of them separately, as if they would not be related.
  2. Line 54. Please, provide explanation of the VGIs abbrviation.
  3. Line 84. Typographical error in "streptococcal", and this word should not be written in italic font.
  4. Line 139. Felix d'Herelle was born in Paris, and his nationality was French, though it is true that he then moved to Canada. Therefore, the most proper statement would be that he was French-Canadian, rather than Canadian.
  5. Line 202. Enterococcus faecium was mentioned in the text previously, thus, E. faecium should be used here.
  6. Line 208. Typographical error - replace "P. aeroginusas” with „P. aeruginosa".
  7. Line 221. Explain the abbreviation MSSA (it is mentioned further in line 229, but without this abbreviated form).
  8. Lines 231-232. Information that 5 mL of bacteriophage lysate was used, without indicating a titer of this lysate, is useless. Please, provide the dose of the phage which was used.

Author Response

  1. Done. We added a paragraph under “2. Pathogenic bacteria and vascular graft infections” to describe the problem of biofilm in a more detailed way (please refer to lines 49-56 and references 6-9).
  2. Done. We amended most common etiological factors of VAD infections. (please refer to lines 69-76 and references 12+13)
  3. Done. We amended the section “5. Bacteriophage therapy” by adding figure 3 and a detailed description on the process of prophage induction with special consideration of Shiga toxin producing E. coli and the role of cholera toxin-encoded bacteriophages. (please refer to lines 163-176) and cited corresponding literature (references 40-45).
  4. Done. We amended more details regarding cases described.

(231-234) Potentially suitable bacteriophage strains were selected from the well-characterized collection housed in the Gabrichevsky Institute. Lysis efficacy was evaluated by serial dilution spot testing and efficiency of plating was analyzed by a double layer plaque assay.

 (238-243) To examine potential synergy or additive effects and exclude concerns of phage/antibiotic antagonism prior to emergency application of phage OMKO1, we performed in vitro biofilm reduction assays using the strain isolated from fistular dis-charge of the patient. Biofilms were grown on 3 mm × 3 mm sections of Dacron by in-oculating each section in 150 µl 0.1 × LB broth in a 96-well dish with 50 µl of an over-night culture of P. aeruginosa isolated from fistular discharge of our patient.

  (249-256) During surgical debridement, no antiseptics were used. Bacteriophages were applied as a loading dose of bacteriophage solution (10 mL) containing 1:1 mixture of PYO bacteriophage at  plaque-forming units (PFU)/mL (R-022600) and Staphylococcal bacteriophage Sb-1at  PFU/mL (R-022876), both from ELIAVA Institute of Bacteriophage, Tbilisi, Georgia. After surgery, a drain was placed into the surgical site before wound closure, through which 5 mL of bacteriophage solution was instilled every 8 h for 14 days. After 21 days, intravenous antibiotics were switched to oral Levofloxacin (500 mg/12 h) and Rifampicin (450 mg/12 h) for 3 months.

 (274-280)The bacteriophage solution (10 ml) was a 1:1 mixture of PYO bacteriophage with plaque-forming units/ml (Georgian pharmaceutical product registration number: R-022600) and Staphylococcal bacteriophage Sb-1 with plaque-forming units/ml (R-022876), both obtained from the Eliava Institute of Bacteriophage, Microbiology and Virology, Tbilisi, GA. Bacteriophage products are a mixture of sterile filtrates of phage lysates which have previously been shown to be active against S. aureus isolates from the patient.

Minor points:

  1. Microbiological nomenclature was corrected. Please refer to lines 45-46.
  2. VGI = Vascular graft infections (abbreviation in line 63 and in abbreviation list included).
  3. Typographical correction of “streptococcal” was performed. (line 99)
  4. Correct designation of Felix d’Herelle was declared. (line 154)
  5. The term E. faecium was used (line 229)
  6. Typographical correction of Pseudomonas aeruginosa was performed. (line 244)
  7. The abbreviation MSSA was added in the further mentioned place. (line 264)
  8. A detailed description of phage titer in applied bacteriophage solution was added. (please refer to lines 274-280).

Round 2

Reviewer 1 Report

Authors have addressed most of the comments and manuscript looks improved now. Fig.#3 when authors reproduce the image/picture from other journal, authors needs to modify according to your requirement, i.e. make slight changes which is more meaningful for your writing. Overall this manuscript is improved a lot. 

Author Response

The figure used has been modified according to our focus of the article. The lysogenic cycle was elaborated more precisely and the term prophage was added.
The labeling of the lytic cycle steps was simplified.

Reviewer 2 Report

Most of previous comments have been addressed. However, a few minor points should still be corrected, as follows:

  1. Line 167 - "stx" should be in italic font (as it indicates a gene).
  2. Line 252 - "Staphylococcal" should be either "staphylococcal" or "Staphylococcus".
  3. Line 276 - "Staphylococcal" should be either "staphylococcal" or "Staphylococcus".
  4. Figure 3 - Description of the switch from lysogenic to lytic development should be indicated as "Induction of prophage...." rather than "Release of prophage...." (the text under the arrow).

Author Response

Appropriate orthographic corrections were made. Please refer to line 167 ("stx"), 252 and 276 ("staphylococcal").

The labeling of the lysogenic cycle steps has been revised. "Induction of prophage...." has replaced the description "Release of prophage....".